# Effective Length Prediction and Pullout Design of Geosynthetic Strips Based on Pullout Resistance

**DOI:** 10.3390/ma14206151

**Published:** 2021-10-16

**Authors:** Jeongjun Park, Gigwon Hong

**Affiliations:** 1Incheon Disaster Prevention Research Center, Incheon National University, Incheon 22012, Korea; smearjun@hanmail.net; 2Department of Civil and Disaster Prevention Engineering, Halla University, Wonju-si 26404, Korea

**Keywords:** geosynthetic, design, pullout resistance, effective length, reinforced earth

## Abstract

In this study, pullout tests were conducted on geosynthetic strips which can be applied to a block-type front wall. Based on the test results, the effective length is predicted, and the pullout design results are presented. In other words, the pullout displacement–pullout load relationship of all geosynthetic strips was analyzed using the pullout test results, and their effective lengths were predicted. It was found that the reinforcement width affected the pullout force for the geosynthetic strips at the same tensile strength. The pullout behavior was evidenced within a range of approximately 0.45 L of the total length of the reinforcement (L) and hardly occurred beyond a certain distance from the geosynthetic strips front regardless of the normal stress. Based on these pullout behavioral characteristics, a method is proposed for the prediction of the effective length (L_E_) and maximum effective length (L_E(max)_) of a geosynthetic strip. The pullout strength was compared using the total area and effective area methods in accordance with the proposed method. In the case of the total area method, GS50W (width: 50 mm) and GS70W (width: 70 mm) exhibited similar pullout strengths. The pullout strength by the effective area method, however, was found to be affected by the soil-reinforcement interface adhesion. The proposed method used for the prediction of the effective length of a geosynthetic strip was evaluated using a design case. It was confirmed that the method achieved an economical design in instances in which the pullout resistance by the effective length (L_E_) was applied compared with the existing method.

## 1. Introduction

Reinforced earth achieves reinforcement with high-tensile strength, and it has been used as a method to improve the stability of various geotechnical structures by decreasing the earth pressure and by increasing the shear strength [1,2,3,4,5,6,7,8,9,10,11,12,13]. Among the various structures, mechanically stabilized earth (MSE) walls are mainly divided in panel and block types, depending on the type of the facing wall. Because the inhibition of horizontal deformation in the facing wall is required for vertical MSE walls using reinforced earth, various reinforcements have been developed. Reinforcements that are applied to MSE walls are classified as inextensible and extensible reinforcements, depending on the material, and extensible reinforcements, such as geogrids and geosynthetic strips, have been mainly used in the late stages. In South Korea, in particular, block-type MSE walls with geogrids have been mainly applied owing to their economic feasibility and appearance, and geosynthetic strips that are applicable to block-type facing walls have been developed.

The design of the MSE walls was based on internal and external stability. To achieve internal stability, fracture and failure are determined by the tensile strength of the reinforcement. The pullout failure of the resistance area is based on the soil-reinforcement interaction, which is very important for the behavior of MSE walls. The mechanism associated with this failure is very complex because it depends on the characteristics of the soil-reinforcement interaction [14]. In particular, because embedded extensible reinforcement may exhibit tensile deformation owing to load transfer during the pullout process, load transfer must be considered in pullout resistance evaluation. Therefore, it is necessary to analyze the pullout behavior of the reinforcement to secure the stability of the MSE walls and to calculate reasonable design parameters. Many studies have been conducted on the reinforcement interaction. Experimental studies have been conducted on the pullout behavior of grid-type inextensible and extensible reinforcements applied in sand and cohesive soil [15,16,17,18,19,20,21,22]. In addition, studies on methods for evaluating the pullout resistance of reinforcement have been conducted based on experimental research [23,24,25], and a number of studies on the theoretical and numerical analysis of the soil-reinforcement interaction have also been conducted [26,27,28,29]. Studies have been conducted on the influence of soil conditions on the pullout resistance of reinforcement [30,31,32]. Additional studies have been conducted on the shear resistance of reinforced soil and on interactions using mixed soil [33,34,35]. Grid-type reinforcement has been used in many studies, but studies on the pullout behavior of geosynthetic strips have been conducted to extend the applicability of geosynthetic strips [27,28,36,37].

In this study, pullout tests on geosynthetic strips were conducted to ensure the design applicability of these strips to block-type facing walls. Based on the test results, the pullout displacement–pullout load relationship was analyzed to evaluate the pullout behavior of the geosynthetic strips. In addition, the effective length that induced the pullout resistance of a geosynthetic strip in the soil was predicted. The pullout parameters were derived using the predicted effective length, and they were applied to a design case to evaluate the effective length prediction method.

## 2. Theoretical Background of Reinforcement Pullout

### 2.1. Soil-Reinforcement Interaction under Pullout Condition

The interaction between the soil and embedded reinforcement consists of two mechanisms: the shear resistance (friction resistance) on the top and bottom surfaces of the reinforcement, and the bearing resistance of the supporting member. The pullout resistance that uses these mechanisms is expressed by Equation (1) by FHWA [38].
(1)$$P_{r}=2\;L_{e}\;\sigma_{v}^{\prime }\;\alpha \;F^{*},$$
where $L_{e}$ is the embedded length in the resistant zone; $\sigma_{v}^{\prime }$ is effective vertical stress at soil-reinforcement interfaces, α is the scale effect correction factor, $F^{*}\left(=f_{b}\cdot tan\emptyset =tan\delta \right)$ is the pullout resistance factor, $f_{b}$ is the soil-reinforcement bond coefficient, ∅ is the internal friction angle of soil, and δ is the soil-reinforcement interface friction angle.

The pullout resistance factor includes both friction and bearing resistance elements. Because geosynthetic strips develop pullout resistance owing to friction, the pullout resistance factor is identical to the friction angle of the soil-reinforcement interface. The pullout resistance factor is determined by the bond coefficient ($f_{b}$, soil-reinforcement bond coefficient) caused by the soil-reinforcement interaction. The bond coefficient is defined as the ratio of the shear strength on the soil-reinforcement interface to the shear strength of the soil, as shown by Equation (2) [33].
(2)$$f_{b}=\frac{\tau_{p}}{\tau }=\frac{c_{p}+\sigma_{v}^{\prime }\;tan\delta }{c+\sigma_{v}^{\prime }\;tan\emptyset },$$
where $\tau_{p}$ is the shear strength at the soil-reinforcement interface, τ is the shear strength of the soil, $c_{p}$ is the soil-reinforcement interface adhesion, and c is the cohesion in the soil.

In this instance, the shear strength at the soil-reinforcement interface can be calculated using the surface area of the reinforcement and the pullout force from the pullout test results. The mechanism of soil-reinforcement interaction can be confirmed in detail through schematic diagrams in previous studies [33,38].

### 2.2. Evaluation Method on Pullout Resistance of Geosynthetic Strip

For inextensible reinforcement, the pullout resistance design parameters can be easily calculated using the shear strength on the soil-inextensible reinforcement interface because there is little change in the contact area between the reinforcement and soil attributed to tensile deformation during the pullout process. However, extensible reinforcement exhibits tensile deformation in normal stress conditions in soil. In other words, the pullout displacement of the reinforcement decreased as the distance from the front increased during the pullout process. Therefore, the effective area (contact area between the reinforcement and soil) is a very important factor for the calculation of the pullout resistance design parameters of extensible reinforcement.

Ochiai et al. [23] proposed a soil-reinforcement shear strength evaluation method using pullout test results to evaluate the pullout resistance of the extensible reinforcement. This evaluation method comprises a mobilizing process method and an average resistance method. The mobilizing process method evaluates the pullout resistance using the difference in tensile forces between two nodes in arbitrary pullout force conditions and is applicable only to grid-type reinforcement. The average resistance method evaluates the pullout resistance by considering the pullout force distribution along the reinforcement length subject to the maximum pullout force condition and based on the use of the average pullout force. The average resistance method is subclassified in the total area, effective area, and maximum slope methods, depending on the average value calculation method. Their evaluation methods were as follows.

First, the total area method assumes that the pullout resistance applies to the entire area of the reinforcement and is expressed in the form of Equation (3).
(3)$$\tau_{av}=\frac{F_{T_{max}}}{2BL},$$
where B and L are the width and length of the reinforcement, respectively, and $F_{T_{max}}$ is the maximum pullout force.

The effective area method assumes that the pullout resistance applies only to the area wherein the tensile deformation of the reinforcement occurs and is expressed in the form of Equation (4).
(4)$$\tau_{av}=\frac{F_{T_{max}-}F_{r}}{2BL_{T}},$$
where $L_{T}$ is the effective length of the reinforcement, and $F_{T_{max}-}F_{r}$ is the effective tensile force corresponding to $L_{T}$.

Finally, the maximum slope method assumes the pullout resistance when the slope of the tangent in the reinforcement-length–tensile-force distribution curve has a maximum value and is expressed according to Equation (5).
(5)$$\tau_{av}={\left(\frac{dF}{dL}\right)}_{\max}$$

The mechanism of each evaluation method can be confirmed in detail through the figures in the previous study [23].

## 3. Pullout Tests

Large-scale pullout tests were conducted to evaluate the pullout behavior of the geosynthetic strips.

### 3.1. Apparatus of Large-Scale Pullout Test

The apparatus for the large-scale pullout test was composed of a rigid (soil) box, a load (normal and pullout) device, and a control box as shown in Figure 1. The rigid box (length: 1600 mm, width: 760 mm, and height: 550 mm) was larger than the minimum recommendations (610 mm length, 460 mm width, and 305 mm height) specified in ASTM D6706-01 [39]. For normal stress, uniformly distributed loading was enabled using an air bag in consideration of the field loading conditions, and up to 500 kN/m^2^ could be applied. For pullout loads, up to 200 kN could be applied using the displacement control method (0.5 to 30 mm/min).

### 3.2. Materials

#### 3.2.1. Soil

The soil samples used in this study were weathered granite soil samples. It is the most extensively distributed soil type in Korea [40] and is a typical nonplastic cohesive soil that contains fine-grained soil. The soil sample was classified as SW based on the USCS. The particle size distribution and engineering properties of the soil are shown in Figure 2 and Table 1, respectively. The model soil for the pullout tests was compacted with an optimum water content of 14.1%, and a maximum dry unit weight of 18.8 kN/m^2^ was achieved. The shear strength of the soil by direct shear tests was 8.7 kPa, and the internal friction angle was 35.6°.

#### 3.2.2. Geosynthetic Strip

As shown in Figure 3, the geosynthetic strips used in the pullout tests consisted of high-strength polyester and polyethylene. The widths of the geosynthetic strips were 50 and 70 mm, respectively, and both had a manufacturing strength of 25 kN. Geosynthetic strips with widths of 50 and 70 mm were used to investigate the pullout resistance characteristics according to the effective area for the same tensile strength condition.

The wide-width tensile test on the geosynthetic strips with different widths was conducted five times, and the averaged results are shown in Figure 4.

For the geosynthetic strip with a width of 50 mm, the maximum tensile strength was found to be 26.8 kN, and the tensile strain was 10.5%. For the geosynthetic strip with a width of 70 mm, the maximum tensile strength and tensile strain were found to be 33.6 kN and 13.3%, respectively. The tensile strength and tensile strain increased as the width increased, and the two geosynthetic strips exhibited similar tensile deformation behaviors. As the tensile strengths obtained from the wide-width tensile test were higher than the manufacturing strength, there was no problem in applying the geosynthetic strips to pullout tests.

### 3.3. Testing Program

Large-scale pullout tests were conducted in accordance with the ASTM D 6706-01 test method [39].

As shown in Figure 5, the model soil was divided into upper and lower parts with respect to the reinforcement and each part was compacted in three layers with the use of a rammer (impact number per min: 640–680) with an impact force of 14 kN. The compaction rate of each layer was higher than 95%. Figure 5 shows the reinforcement installed in the soil and the deformation measurement positions. The deformation measurement positions were selected to evaluate the effective length through the tensile strain and pullout force distribution in the reinforcement based on previous studies [23,28,30]. In all cases, the pullout deformation was controlled with a strain rate of 1 mm/min, and a backfill height of approximately 8.0 m was applied for the maximum normal stress. In addition, lubrication was performed using wrap and oil to minimize the influence of the wall friction of the soil box.

In the testing program, different horizontal spacings (260 and 210 mm) were applied for the installation of the two types (50 and 70 mm widths) of the geosynthetic strips based on considerations of the applied block-type facing wall, as shown in Table 2. In this instance, when the horizontal spacing was 210 mm, pullout tests were conducted only in the normal stress condition (100 kPa). This corresponded to a backfill height of approximately 5 m in consideration of the most extensively applied MSE wall height in Korea for examination of the influence of the horizontal spacing.

## 4. Test Results and Analysis

### 4.1. Test Results

Based on the pullout force–pullout displacement relationship from the pullout test results, the pullout strength characteristics of the reinforcement can be evaluated using the maximum pullout force at each normal stress condition. In this study, the maximum pullout force for pullout strength evaluation was selected by referring to the displacement criteria suggested in the FHWA design criteria [38]. In other words, when the end displacement of the reinforcement installed in soil was less than 15 mm, it was necessary to determine the development of the maximum pullout force. This means that the pullout force corresponding to an end displacement of 15 mm must be applied as the maximum pullout force if the end displacement of the replacement exceeds 15 mm in the case in which the maximum pullout force was developed. Therefore, as shown in Figure 5b, the measurement position of 110 m from the front sides of the linear variable differential transformers installed in the reinforcement was applied as the end displacement.

In the cases of normal stress conditions of 50, 100, and 150 kPa, the maximum pullout forces were found to be 14.6, 20.4, and 26.8 kN for GS50W, and 19.3, 28.2, and 34.3 kN for GS70W, respectively.

Considering the FHWA design criteria [38], the relationships between the pullout force and front/end displacement according to the width of the geosynthetic strip and the horizontal spacing of reinforcement are shown in Figure 6 and Figure 7. In all experimental cases, the maximum pullout force was observed when the front displacement ranged between 25 and 35 mm (Figure 6). Specifically, the maximum pullout force was approximately 5 to 8 kN higher when the width was 70 mm compared with a width of 50 mm. This is similar to the results of the wide-width tensile test. The maximum pullout force generated at the end displacement also satisfied the FHWA design criteria [38] (Figure 7).

From the pullout force–pullout displacement relationship at a normal stress of 100 kPa, the pullout force–front displacement relationship and the pullout force–end displacement relationship exhibited similar behaviors regardless of the horizontal spacing of the reinforcement. In other words, it was confirmed that friction-resistance reinforcement with the same tensile strength exhibited the same pullout resistance and similar pullout behavior if the installation areas per unit width were identical. Therefore, the analysis used for the prediction of the pullout behavior and effective length of the reinforcement was conducted using only the test results with a larger horizontal spacing (260 mm, GS50W, and GS70W).

### 4.2. Pullout Behavior of Geosynthetic Strip

Figure 8a–c or Figure 9a–c show the pullout displacements as a function of the distance from the reinforcement front. The front pullout displacement was applied to a maximum of 60 mm after the maximum pullout force was developed. As the front pullout displacement increased, the displacement at each position showed a tendency to increase regardless of the reinforcement width. The displacement difference increased as the distance from the reinforcement front approached zero. This tendency was more obvious when the reinforcement width was wider (GS70W) and during exposure to higher normal stress conditions. Therefore, it was found that the pullout force transmitted to the reinforcement in soil during the pullout process was maximized at the front and decreased as the distance from the front increased. It was also found that the pullout force induced by the increase in normal stress was concentrated at the front.

Figure 8d or Figure 9d show the pullout displacement according to the distance from the reinforcement front under the maximum pullout force condition. As shown in the figures, the pullout displacement decreased as the normal stress increased. At positions near the front, the reduction rate of the displacement increased at normal stress conditions. However, when the distance from the front exceeded approximately 45% to 50% of the entire reinforcement length, the reduction rates were found to be similar.

Figure 10 shows the results of analyzing the pullout force induced in the geosynthetic strips at the maximum pullout condition using the pullout displacement evaluation results according to the distance from the reinforcement front. The distance from the front was applied as the ratio of the displacement measurement position length (L_i_) to the total length of the reinforcement (L).

It was found that the pullout force induced in the geosynthetic strips occurred for a length ratio (L_i_/L) of approximately 0.45. In addition, it was found that 57–75% of the pullout force was induced at the front in the case of GS50W, and 68–83% in the case of GS70W, depending on normal stress conditions. For cases GS50W and GS70W, similar pullout forces were induced subject to the length ratio conditions of 0.5 L and 0.47 L, respectively. Furthermore, there was little change in the induced pullout force after 0.75 L and 0.73 L, at which the tensile strain values of GS50W and GS70W (calculated based on Figure 8 and Figure 9) were approximately 1%.

### 4.3. Prediction of Effective Length Considering Pullout Force

The pullout force distribution in the geosynthetic strip occurred within a limited reinforcement length range. Therefore, it will be possible to achieve a more economical design if the effective length at which the pullout resistance is induced by the pullout force is calculated. The relationship between the pullout force and L_i_/L (as shown in Figure 10) can be simplified as shown in Figure 11, and it is possible to be used to reflect the experimental results. As mentioned previously, in the cases of GS50W and GS70W, the pullout forces of the reinforcement at normal stress conditions became almost similar at a length ratio (L_i_/L) of approximately 0.5 L. Subsequently, the induced pullout force decreased and then hardly changed after 0.75 L, at which the strain became approximately 1%. Therefore, the effective length of the geosynthetic strips used in this study was determined as follows: the effective length (L_E_ = 0.5 L) was defined as the length at which the pullout force became almost similar, regardless of the normal stress, and the maximum effective length (L_E(max)_ = 0.75 L) as the distance at which the pullout force hardly changed.

### 4.4. Evaluation of Pullout Resistance Considering the Prediction of Effective Length

To evaluate the effective length prediction method presented in Section 4.3, the pullout strength was evaluated according to the effective length (L_E_) and maximum effective length (L_E(max)_), respectively, based on the use of the average resistance method proposed by [23]. The pullout strength was evaluated using the total and effective area methods based on consideration of the extensibility of the geosynthetic strips, and the results are shown in Figure 12 and Table 3.

Regarding the pullout strength of GS50W based on considerations of the maximum effective length (L_E(max)_), the results of the effective area method and total area method were similar at low normal stress conditions because the pullout force was transmitted to the maximum effective length (L_E(max)_). However, as the normal stress increased, the pullout strength gradually increased because the pullout force transmitted to the maximum effective length (L_E(max)_) decreased. For GS50W, the effective area method that considered the effective length (L_E_) yielded a higher pullout strength in normal stress conditions compared with the total and the effective area methods that considered the maximum effective length (L_E(max)_) because the distance from the front decreased. The same tendency was observed for GS70W.

The pullout strengths of GS50W and GS70W with respect to the reinforcement width were compared. GS50W and GS70W exhibited similar pullout strengths based on the total area method. The pullout strength obtained by the effective area method, however, showed that the reinforcement width had an influence on the soil-reinforcement interface adhesion. This was more obvious when the effective length (L_E_) was applied.

Figure 13 shows the calculation results of the bond coefficient of the geosynthetic strips using the shear strength of the soil and the pullout strength ratio of the soil-reinforcement interface. For all evaluation methods, the bond coefficient slowly decreased and then exhibited a tendency to converge as the normal stress increased. Tatlisoz et al. [33] reported that the soil-reinforcement interface adhesion was sufficient when the bond coefficient was 1.0, and the adhesion was low when the bond coefficient was 0.5 or less. For GS50W and GS70W, the bond coefficient was the lowest when the normal stress was 150 kPa; the respective values were 0.88 and 0.80 in the case of the total area method, respectively. These values are higher than the friction coefficient of soil (0.71) and are, thus, considered stable. Therefore, for a more efficient design of geosynthetic strips, the effective length (L_E_) and maximum effective length (L_E(max)_) can be considered. However, it is reasonable to apply the effective length (L_E_) because there is no significant difference in the pullout force between the effective length (L_E_) and maximum effective length (L_E(max)_).

### 4.5. Design Case on Pullout Resistance Based on Considerations of the Effective Length

The effective length prediction method and bond coefficient were used and applied to the design case. The height of the block-type MSE wall applied to the design case was set to 7.8 m, which is close to the standard section of the block-type MSE wall in Korea. In this instance, the bond coefficient was set to the minimum value, which corresponded to the normal stress condition at 150 kPa to secure the stability of the structure, and the reinforcement installation length was set to 5.46 m (Figure 14). The FHWA design criteria [38] were applied to the design method.

Based on the proposed pullout resistance design method, the effective length of the reinforcement that satisfied the minimum stability for the internal and external stability of the MSE wall was calculated, and the results are listed in Table 4. Because all conditions were identical except for the bond coefficient, the reinforcement length (L_a_) in the active zone was identical. However, when the effective length prediction method was applied to the pullout resistance evaluation method, the effective length and total length of the reinforcement in the resistant zone were calculated differently. In other words, the total area method exhibited the largest effective lengths for GS50W and GS70W, which satisfied the external and internal stability of the MSE wall, followed by L_E(max)_ and L_E_. This means that the design pattern that applied the total area method to the design of the pullout resistance of extensible reinforcement underestimated the stability of the structure. Therefore, the application of the effective area method that considered the effective length can lead to economical designs.

The effective length of GS70W was shorter than that of GS50W in the resistant zone because it had a higher area ratio owing to the reinforcement width. Therefore, it is possible to secure stability even if the total length of the reinforcement was reduced by 0.5 to 1 m based on the effective area evaluation method.

L_E_ can be applied to the design of the pullout resistance by considering the effective length of the geosynthetic strip. This method can satisfy both economic feasibility and stability. In the case of the application of weathered granite soil, which is generally used as backfill soil, it was confirmed that L_E_ and L_E(max)_ took values that ranged from approximately 0.5 L to 0.75 L. Given that the pullout force generated between 0.5 L and 0.75 L had no significant influence on the stability of the MSE wall, the pullout resistance design method that used L_E_ based on the proposed method was sufficiently applicable.

## 5. Conclusions

In this study, the effective length of reinforcement was predicted by considering the pullout force distribution using the indoor pullout test results for the geosynthetic strips. In addition, the applicability of the predicted effective length was evaluated using a design case. The analyzed results were as follows:(1)The geosynthetic strips with the same tensile strength exhibited similar pullout behaviors regardless of the horizontal spacing of reinforcement, but the reinforcement width had a more significant influence on the tensile strength and pullout force than the horizontal reinforcement spacing.(2)The pullout behavior of the geosynthetic strips was concentrated at the front, and this was more obvious as the normal stress increased. Beyond a certain distance from the front, however, the pullout behavior of the reinforcement hardly occurred regardless of the normal stress. Based on this, it was possible to propose a method to predict the effective length (L_E_) and maximum effective length (L_E(max)_).(3)The pullout resistance estimated based on the predicted effective length showed results that could secure stability. In addition, it was found that the effective area method—that considered the prediction of the effective length based on the use of the pullout test geosynthetic strip results—was applicable as an efficient pullout resistance design method. In other words, the proposed method was found to be more suitable for economical designs than the existing method based on the results of the design case that used the effective length for the pullout resistance of the geosynthetic strips.

In this study, soils which has a specific particle size distribution and optimum water content were applied to the pullout test. The water content and particle size distribution of soils may affect pullout resistance, effective length prediction, and pullout design because the soil conditions applied in this study are limited. Therefore, it is necessary to study the pullout resistance applied to various soil conditions.

## Figures and Tables

**Figure 1 materials-14-06151-f001:**
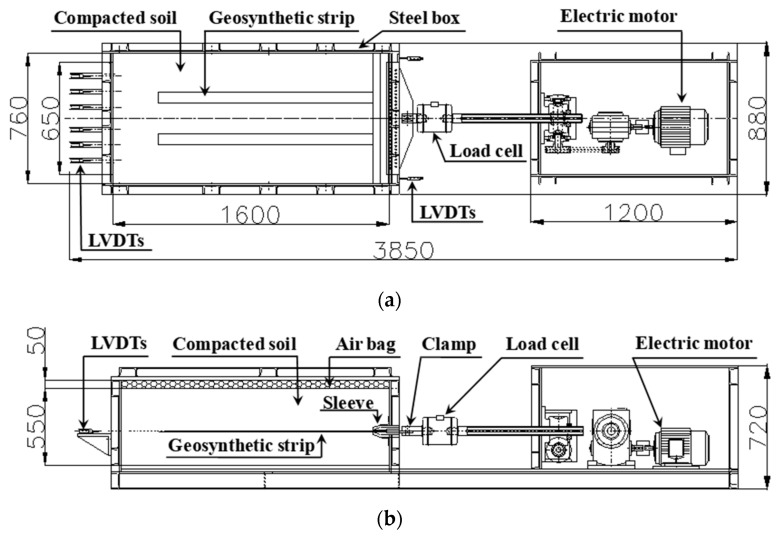
Schematic of large-scale pullout test apparatus: (**a**) plan view; (**b**) cross-sectional views (dimensions in millimeters).

**Figure 2 materials-14-06151-f002:**
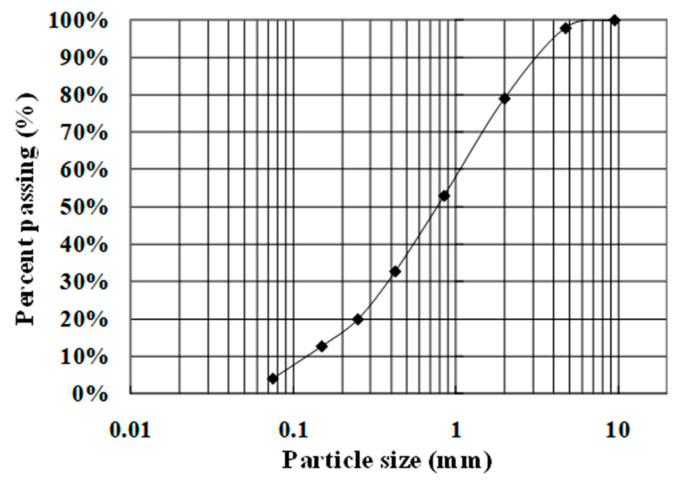
Particle size distribution of soil.

**Figure 3 materials-14-06151-f003:**
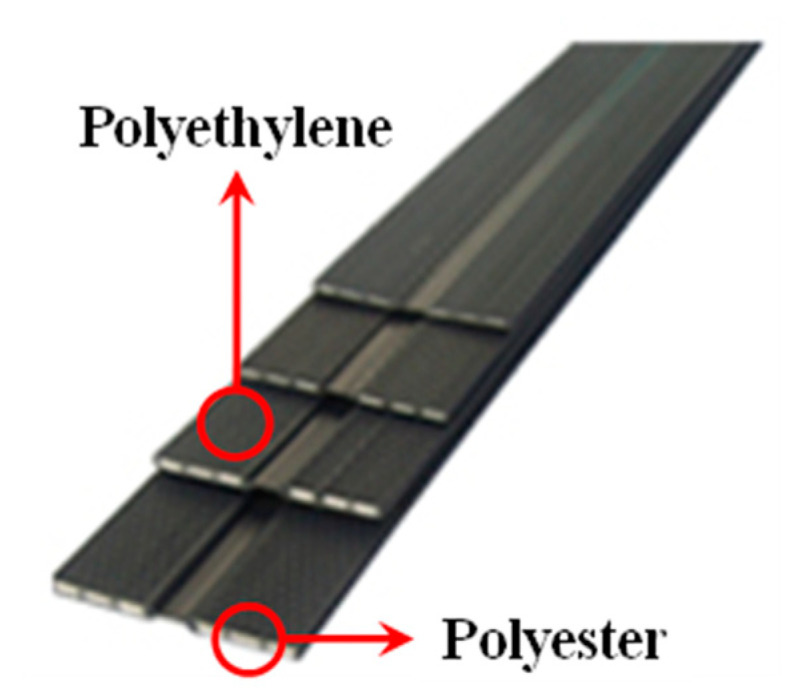
Composition of geosynthetic strip.

**Figure 4 materials-14-06151-f004:**
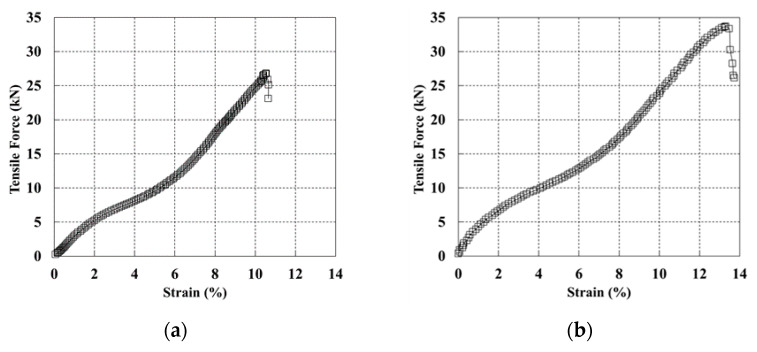
Relationship between tensile force and strain on the reinforcement with widths equal to: (**a**) width 50 mm; (**b**) width 70 mm.

**Figure 5 materials-14-06151-f005:**
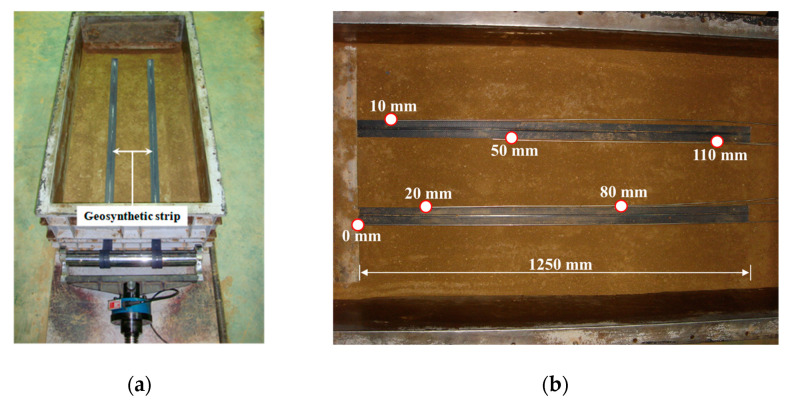
View of the installed geosynthetic strip: (**a**) setup view; (**b**) measurement locations.

**Figure 6 materials-14-06151-f006:**
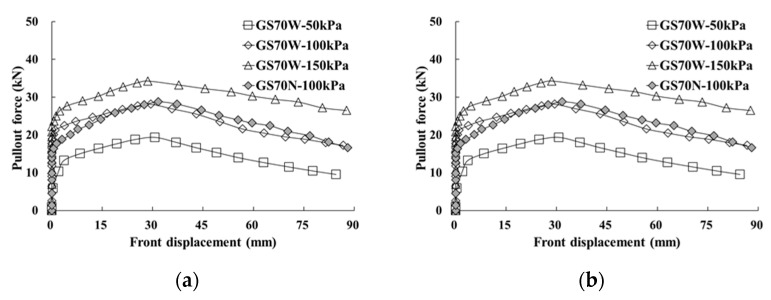
Relationship between pullout force and front displacement for widths equal to: (**a**) 50 mm; (**b**) 70 mm.

**Figure 7 materials-14-06151-f007:**
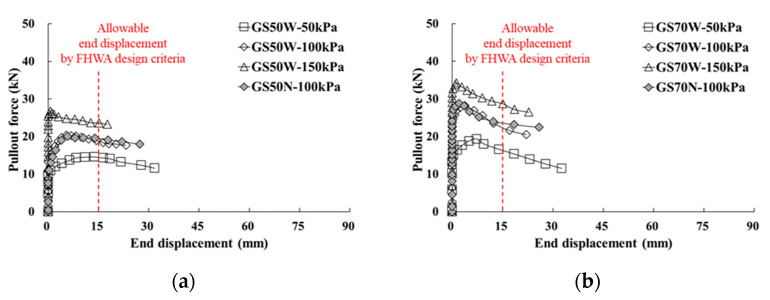
Relationship between pullout force and end displacement for widths equal to: (**a**) 50 mm; (**b**) 70 mm.

**Figure 8 materials-14-06151-f008:**
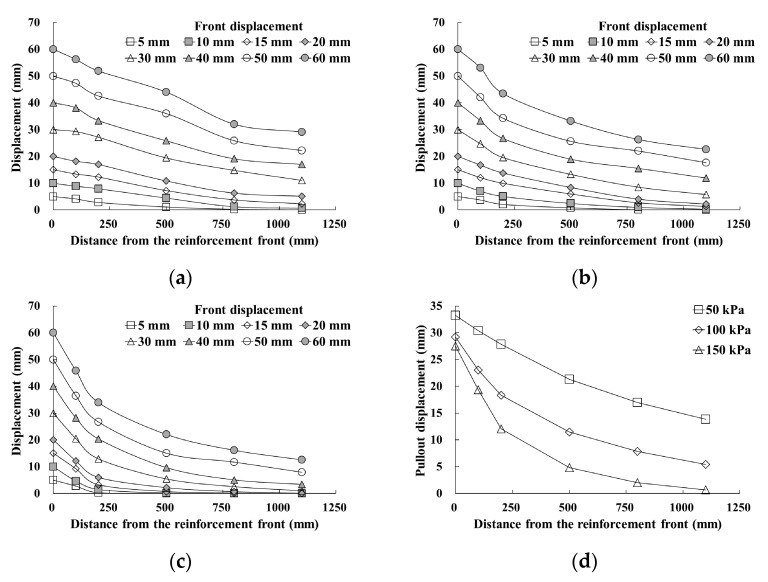
Displacement as a function of distance from the geosynthetic strip front: (**a**) GS50W-50 kPa; (**b**) GS50W-100 kPa; (**c**) GS50W-150 kPa; (**d**) pullout displacement of reinforcement in maximum pullout force.

**Figure 9 materials-14-06151-f009:**
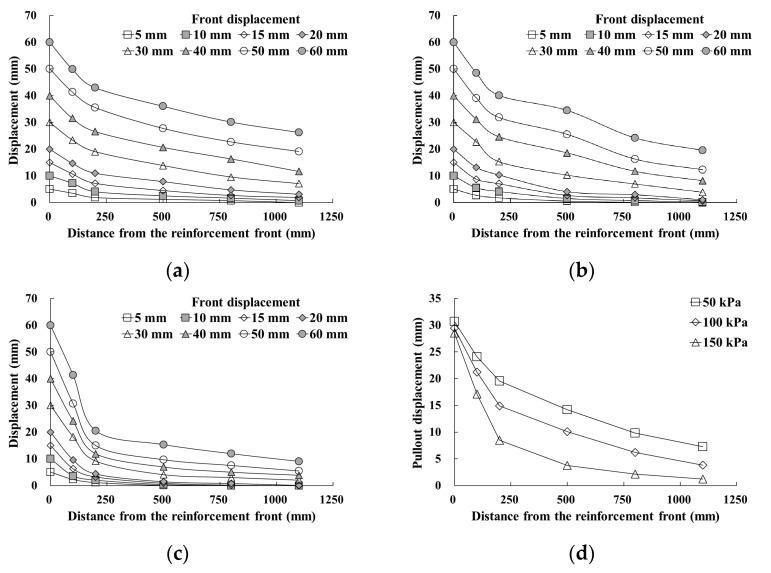
Displacement as a function of distance from the geosynthetic strip front: (**a**) GS70W-50 kPa; (**b**) GS70W-100 kPa; (**c**) GS70W-150 kPa; (**d**) pullout displacement of reinforcement in maximum pullout.

**Figure 10 materials-14-06151-f010:**
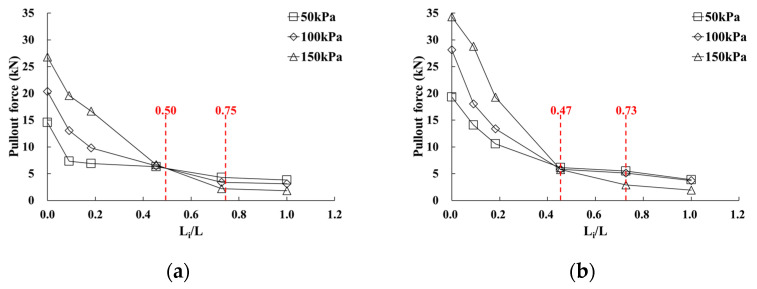
Relationship between pullout force and L_i_/L according to reinforcement width: (**a**) GS50W; (**b**) GS70W.

**Figure 11 materials-14-06151-f011:**
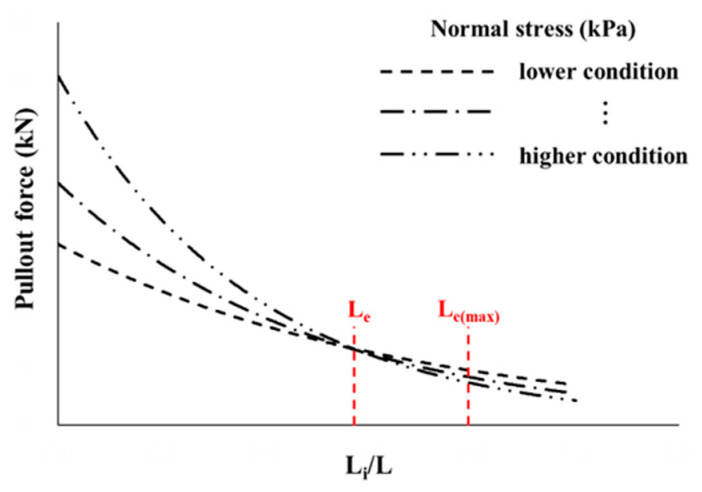
Prediction of effective length using the pullout force generated in reinforcement cases.

**Figure 12 materials-14-06151-f012:**
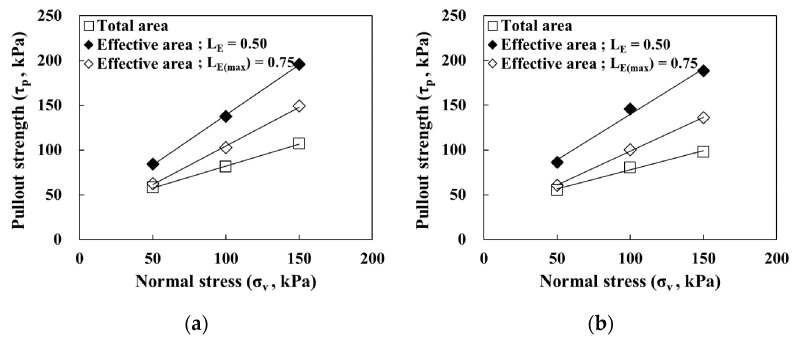
Relationship between normal stress and pullout strength: (**a**) GS50W; (**b**) GS70W.

**Figure 13 materials-14-06151-f013:**
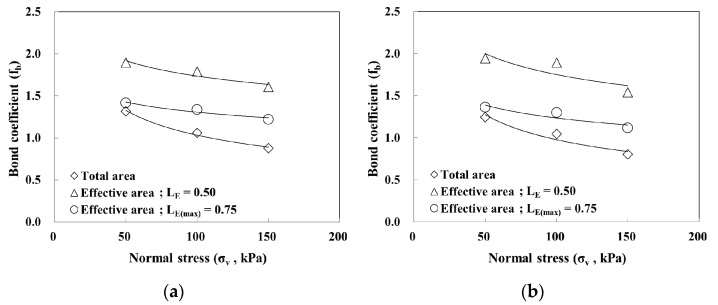
Relationship between normal stress and bond coefficient: (**a**) GS50W; (**b**) GS70W.

**Figure 14 materials-14-06151-f014:**
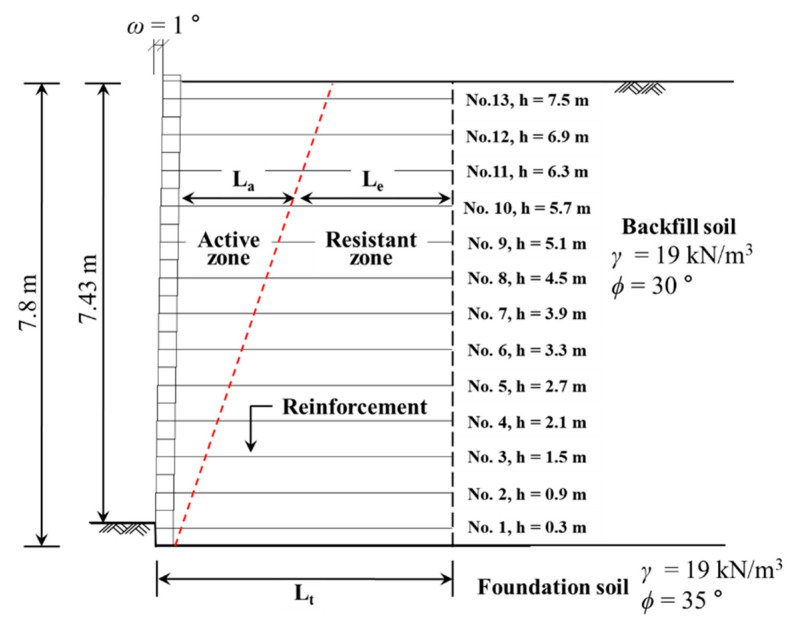
Cross-sectional view of design case.

**Table 1 materials-14-06151-t001:** Engineering properties of weathered granite soil.

Specific Gravity (*G_S_*)	Plastic Limit (*w_P_*, %)	Maximum Dry Unit Weight (*γ_d,max_*, kN/m^3^)	Optimum Water Content (*w**_op_*, %)	Friction Angle (*Φ*, °)	Cohesion (*c*, kPa)	USCS
2.67	NP (Nonplastic)	18.8	14.1	35.5	8.7	SW (Well-graded sand)

**Table 2 materials-14-06151-t002:** Testing program of geosynthetic strip pullout.

Reinforcement Width (mm)	Horizontal Spacing of Reinforcement (mm)	Normal Stress (*σ_v_*, kPa)	Test Classification
50	260	50, 100, 150	GS50W
	210	100	GS50N
70	260	50, 100, 150	GS70W
	210	100	GS70N

**Table 3 materials-14-06151-t003:** Summary of pullout parameters.

Classification Tests	Pullout Resistance Evaluation Methods	Pullout Parameter	
		Soil-Reinforcement Interface Friction Angle ($c_{p}$, °)	Soil-Reinforcement Interface Friction Angle (δ, °)
50	Total area	33.6	26.0
	Effective area (L_E_)	27.4	48.2
	Effective area (L_E(max)_)	18.6	40.8
70	Total area	35.1	23.1
	Effective area (L_E_)	38.1	45.6
	Effective area (L_E(max)_)	23.2	37.1

**Table 4 materials-14-06151-t004:** Design results according to pullout resistance methods (dimensions in m).

Reinforcement No.	EmbbededHeight (m)	L_a_, Active Zone (m)	L_e_, Resistant Zone (m)						Stability on Pullout Resistance
			GS50W			GS70W			
			Total Area Method	Effective Area Method		Total Area Method	Effective Area Method		
				L_E_ = 0.5	L_E(max)_ = 0.75		L_E_ = 0.5	L_E(max)_ = 0.75	
1	0.3	0.168	6.332	5.332	5.732	5.832	5.292	5.332	Safety factor (FS_po_)on L_e_ of geosynthetic strip ≥ 1.5
2	0.9	0.504	5.996	4.996	5.396	5.496	4.956	4.996	
3	1.5	0.840	5.660	4.660	5.060	5.160	4.620	4.660	
4	2.1	1.176	5.324	4.324	4.724	4.824	4.284	4.324	
5	2.7	1.512	4.988	3.988	4.388	4.488	3.948	3.988	
6	3.3	1.848	4.652	3.652	4.052	4.152	3.612	3.652	
7	3.9	2.184	4.316	3.316	3.716	3.816	3.276	3.316	
8	4.5	2.520	3.980	2.980	3.380	3.480	2.940	2.980	
9	5.1	2.855	3.645	2.645	3.045	3.145	2.605	2.645	
10	5.7	3.191	3.309	2.309	2.709	2.809	2.269	2.309	
11	6.3	3.527	2.973	1.973	2.373	2.473	1.933	1.973	
12	6.9	3.863	2.637	1.637	2.037	2.137	1.597	1.637	
13	7.5	4.199	2.301	1.301	1.701	1.801	1.261	1.301	

## Data Availability

Data presented in this study are available on request from the corresponding author. The data are not publicly available due to data that are also part of an ongoing study.

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
