# Peer review of "Effective Length Prediction and Pullout Design of Geosynthetic Strips Based on Pullout Resistance"

_materials, 2021, doi:10.3390/ma14206151_

Round 1
Reviewer 1 Report
The manuscript presents interesting information and data, and should be suitable for publication following relatively minor revisions. Technically the manuscript is sound. My detailed editorial comments and suggestions are:
1) Title: The title could be shortened to something like “Effective Length and Pullout Design of Geosynthetic Reinforcing Strips”.
2) Abstract: instead of explaining something one way then saying or in other words, you should say what you mean using whichever of two explanations is most clear and precise. This principle applies in any circumstance. In line 19 you should say is proposed rather than was proposed. Again the writing principle (correct choice of tense) applies generally. I’m sure the journal production staff will work with you to correct such things.
3) Keywords: Use simple words rather than compound terms or phrases. For example: design, earth, geosynthetic, pullout, reinforcement.
4) Introduction: The Introduction needs to be edited to remove excessively length sentences. For example, the first sentence contains 49 words, and could be rewritten as a sentence defining the attributes of reinforced earth, and a sentence illustrating uses of it. In general, it is important to not make sentences so complex that readers will need to read them more than once to understand what you want to tell them.
5) Theoretical background: The equations in Section 2.1 would be easier to understand were you to add a schematic diagram of the physical situation to which they reply. Ditto Section 2.2.
6) Pullout tests, Results, Conclusions: These sections seem straightforward, and I therefore have no particular comments on them. The quality and readability of some of the figures might be something the journal production staff will ask you to address.
Reviewer 2 Report
The manuscript deals with pull-out tests on geosyntethic strips to improve stability of retaining walls. The manuscript is well written and the tests are correctly described. Few comments are in the annexed file
